# Quantitative cellular-resolution map of the oxytocin receptor in postnatally developing mouse brains

Kyra T. Newmaster [1], Zachary T. Nolan [1], Uree Chon [1], Daniel J. Vanselow [1,2], Abigael R. Weit [1], Manal Tabbaa[3], Shizu Hidema[4,5], Katsuhiko Nishimori[4,6], Elizabeth A. D. Hammock [3] & Yongsoo Kim [1]✉

The oxytocin receptor (OTR) plays critical roles in social behavior development. Despite its significance, brain-wide quantitative understanding of OTR expression remains limited in postnatally developing brains. Here, we develop postnatal 3D template brains to register whole brain images with cellular resolution to systematically quantify OTR cell densities. We utilize fluorescent reporter mice ($Otr^{venus/+}$) and find that cortical regions show temporally and spatially heterogeneous patterns with transient postnatal OTR expression without cell death. Cortical OTR cells are largely glutamatergic neurons with the exception of cells in layer 6b. Subcortical regions show similar temporal regulation except the hypothalamus and two hypothalamic nuclei display sexually dimorphic OTR expression. Lack of OTR expression correlates with reduced dendritic spine densities in selected cortical regions of developing brains. Lastly, we create a website to visualize our high-resolution imaging data. In summary, our research provides a comprehensive resource for postnatal OTR expression in the mouse brain.

[1] Department of Neural and Behavioral Sciences, Penn State University, Hershey, PA, USA. [2] Department of Pathology, College of Medicine, Penn State University, Hershey, PA, USA. [3] Department of Psychology and Program in Neuroscience, Florida State University, Tallahassee, FL, USA. [4] Tohoku University Graduate School of Agricultural Science, Miyagi, Japan. [5] Present address: Department of Bioregulation and Pharmacological Medicine, Fukushima Medical University, Hikarigaoka 1, Fukushima City, Fukushima Prefecture, Japan. [6] Present address: Department of Obesity and Internal Inflammation, Fukushima Medical University, Hikarigaoka 1, Fukushima City, Fukushima Prefecture, Japan. ✉email: yuk17@psu.edu

Oxytocin receptor (OTR) mediates oxytocin (OT) signaling that plays a critical role in the development of social behavior for animals, including humans[1–3]. Animal models lacking functional OTR show social behavior impairment[4,5], suggesting that OTR expression is important for normal social behavior. OTR expression begins early in life with peak cortical OTR expression coinciding with critical postnatal developmental windows for social learning[2,6,7]. This transient OTR expression in the developing cortex is thought to play an important role in facilitating neural circuit maturation[8,9]. For instance, OTR in postnatally developing brains has been implicated in multisensory binding[10], maturation of GABAergic neurons[11], and synapse formation and maturation between neurons[12,13].

During the early postnatal period and adulthood, many different brain regions contain OTR-expressing cells that are either excitatory or inhibitory neurons[6,12,14,15]. OTR expressing neurons in different brain regions have been linked to neural circuit specific functions, such as facilitating social reward in the ventral tegmental area[16], social recognition in the anterior olfactory nucleus (AON)[17], and social memory in the hippocampal CA2 region[18,19]. However, we have limited knowledge on the temporal and regional expression patterns of OTR throughout the entire brain. Previous studies investigating OTR expression mainly utilized receptor autoradiography binding assays, histological methods (e.g., immunohistochemistry using specific antibodies), or transgenic reporter animals[6,15,20,21]. Most of these studies, if not all, examined selected brain regions by histological methods, which is difficult to apply to whole brain analysis across developmental periods due to variable staining results, laborious procedures, and semiquantitative assessment.

To overcome this issue, we develop new postnatal template brains at different postnatal (P) developmental periods (P7, 14, 21, and 28) with detailed anatomical labels based on Allen Common Coordinate Framework (CCF)[22]. Then, we expand our existing quantitative brain mapping platform (qBrain)[23] to image, detect, and quantify fluorescently labeled cells at the cellular resolution from postnatally developing brains (developmental qBrain; dqBrain). We apply this method to quantify the number and density of OTR (+) cells using Otr-Venus knock-in reporter mice ($Otr^{venus/+}$) after confirming its faithful representation of endogenous OTR expression using fluorescent in situ hybridization[20]. We find temporally and spatially heterogeneous cortical and subcortical OTR expression with peak densities during early postnatal periods, which is important to maintain dendritic spine density. Our cumulative labeling reveals that cortical OTR reduction into adulthood is mainly driven by receptor downregulation without cell death. Furthermore, we identify sexually dimorphic OTR expression in two hypothalamic nuclei. Lastly, we deposit postnatal template brains and high-resolution image data with user friendly visualization in our website (http://kimlab.io/brain-map/OTR/) to facilitate open data sharing.

## Results

### Choice of fluorescent reporter mice for OTR expression.
To quantify OTR expression across the whole brain, we used two transgenic reporter mice that express fluorescent reporters under the OTR promotor. The lines we examined include a BAC transgenic Otr-eGFP reporter mouse[24], and a knock-in $Otr^{venus/+}$ heterozygote mouse (Otr-Venus) that encodes a fluorescent reporter gene (Venus) in place of the genomic Otr coding region[20]. We initially observed significant discrepancies in the number and location of cells reporting OTR expression between the two mouse lines (Fig. 1). In order to independently validate these observations, we used single-molecule mRNA fluorescent

in situ hybridization against Otr in postnatally developing mouse brains. We first confirmed the specificity of the Otr in situ hybridization by comparing expression of Otr mRNA in wild type (WT; $n = 3$ mice, Fig. 1b, e, i, l, p, s) and littermates Otr knockout (KO) mice ($Otr^{venus/venus}$; $n = 2$ mice, Fig. 1w, x) at P21. $Otr^{venus/venus}$ mice expressed no Otr mRNA, whereas their WT littermates showed robust expression. Then, we compared our Otr in situ hybridization results (WT; $n = 3$ mice) to fluorescent reporter expression from both Otr-Venus ($n = 10$) and Otr-eGFP mice ($n = 10$) at P21. We found that Venus expression from Otr-Venus mice overall matched to endogenous OTR expression very closely while OTR-eGFP often lacked comparable expression (false negative) or misrepresented OTR expression (false positive) in several brain regions (Fig. 1a–u). For example, the prelimbic cortex (PL) and the taenia tecta (TT) showed distinct OTR expressions in both Otr in situ and Otr-Venus while very little expression in Otr-eGFP mice (Fig. 1b–d, e–g). We also observed that GFP-labeled cells in Otr-eGFP mice were mostly restricted to the superficial layer of the somatosensory cortex, while Otr-Venus mice showed a population of Venus-labeled cells in both superficial and deep layer that corresponded to the RNA in situ results for analogous areas (Fig. 1i–k). In the posterior cortical area, the Otr-Venus mice exhibit OTR expression that is well matched to our Otr in situ data, while Otr-eGFP reports little expression in the layer 2 of the visual cortex (white arrows in Fig. 1p–r). RNA in situ data also shows that Otr is strongly expressed in the bed nucleus of stria terminalis posterior interfascicular division (BSTif), which is correctly reported by the Otr-Venus reporter (Fig. 1l, m). However, Otr-eGFP mice report GFP expression in the BST posterior principal nucleus (BSTpr), not in the BSTif (Fig. 1n). Moreover, robust OTR expression in the posteromedial cortical amygdala (COApm) was observed in both the Otr in situ data and the Otr-Venus reporter, while little expression was found in the Otr-eGFP reporter (Fig. 1s–u). We further analyzed OTR-Venus expression at adult stage (at P56) in relation to Otr mRNA expression data from publicly available in situ database from Allen Institute for Brain Science[25] and confirmed comparable expression patterns in Otr-Venus mice (Supplementary Fig. 1). We then examined whether the Venus mRNA and Otr mRNA were co-expressed in the same cells from Otr-Venus mice by using double fluorescent in situ hybridization (Fig. 1v). We confirmed that the majority of Venus-positive cells also express Otr mRNA (83.8%, 321 Otr-positive cells among 383 Venus-positive cells in the cortex, the amygdala, and the hippocampus, $N = 3$ mice at P21).

Collectively, we concluded that the Otr-Venus mice can serve as a good reporter line to examine the developmental trajectory of the OTR expression.

### Quantitative mapping in postnatally developing brains.
We previously established a quantitative brain mapping method (termed "qBrain") that can count the number and density of fluorescently labeled cells in over 600 different anatomical regions across the entire adult mouse brain with cellular resolution precision[23]. The method consists of whole brain imaging at cellular resolution using serial two-photon tomography, machine learning-based algorithms to detect fluorescently labeled cells, image registration to a reference brain, and statistical analysis (Fig. 2a–d). To extend the method to map signals in postnatally developing brains, we established registration template brains from different early postnatal periods: P7, 14, 21, and 28 (Fig. 2g–j)[26]. First, we chose the best-imaged brain at each age (Supplementary Fig. 2). Then, we registered brains from the same age to the initial template brain. Lastly, we averaged all transformed brains to generate an averaged template brain at each age

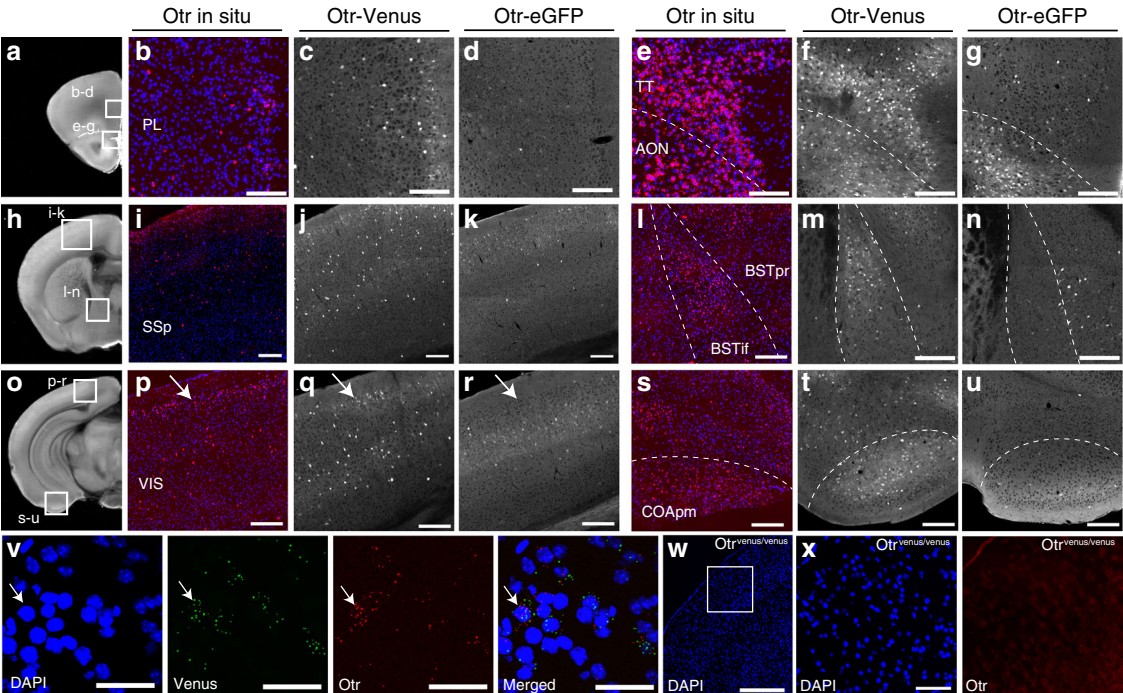

**Fig. 1 Characterization of *Otr* transgenic reporter mice. a–u** Comparison between the *Otr* fluorescent in situ hybridization and *Otr* transgenic reporter mouse lines at P21. Scale bar = 200 μm. The white boxes in the first column represent brain regions in zoomed-in pictures on subsequent columns. The second and the fifth column for the *Otr* in situ, the third and the sixth for the *Otr-Venus* mice, and the fourth and the seventh for the *Otr-eGFP* mice. **b–d** the prelimbic cortex (PL), **e–g** the taenia tecta (TT) and the anterior olfactory nucleus (AON), **i–k** the primary somatosensory cortex (SSp), **l–n** the bed nucleus of stria terminalis (BST) interfascicular (if) and principal (pr) nucleus, **p–r** the visual cortex (VIS), and **s–u** the cortical amygdala posterior medial (COApm) area. Note the corresponding patterns between the *Otr* in situ and the OTR-Venus, but not OTR-eGFP. **v** Double fluorescent in situ against the *Otr* and the *Venus* in the cortex from the *Otr-Venus* mice. The white arrows indicate an example of both *Otr* and *Venus*-positive cells. Scale bar = 50 μm. **w, x** *Otr* in situ hybridization on OTR KO (OTR^*venus/venus*) mice. **x** the somatosensory cortex from the white boxed area in **w**. Note the lack of *Otr* puncta. Scale bar = 400 μm for **w** and 100 μm for **x**.

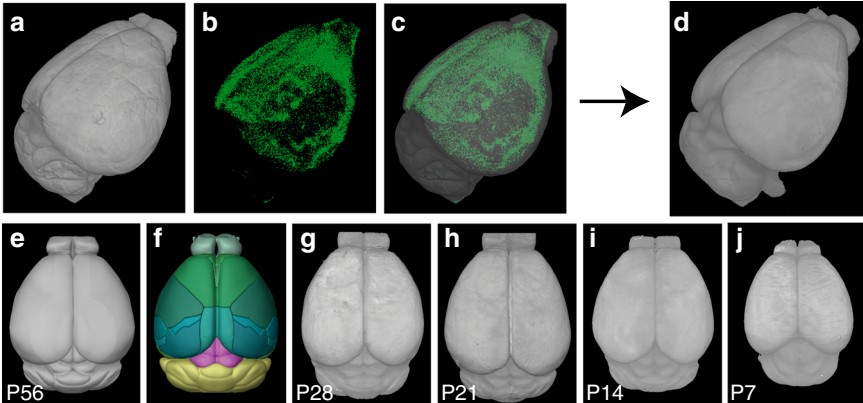

**Fig. 2 Quantitative brain mapping to examine OTR expression in developing postnatal mouse brains. a–c** Reconstructed 3D brains from serial two-photon tomography imaging of the P14 *Otr-Venus* mouse brain **a**, detected Venus-positive cells **b**, and their overlay **c**. **d** The registration template brain for automated cell counting for P14 brains. **e–i** Template brains at different postnatal ages. **e** The adult Allen CCF background template and **f** its anatomical labels. Newly generated template brains at P28 **g**, P21 **h**, P14 **i**, and P7 **j**.

($N$ = 8 brains at P7, 15 at P14, 12 at P21, and 17 at P28). Furthermore, we generated age-matched anatomical labels by transforming labels from the adult brain based on the CCF from Allen Institute for Brain Science to template brains of younger ages (Supplementary Fig. 2). With these tools, termed "dqBrain", we were able to register our image data to age-matched template brains and quantify fluorescently labeled cells across the entire brain at different postnatal ages (Fig. 2, Supplementary Movies 1–5).

**Developmental expression patterns in the isocortex**. To examine regional and temporal heterogeneity of OTR expression, we imaged *Otr-Venus* mice at five different postnatal days (P7, 14, 21, 28, and 56, $N$ = 5 males and 5 females per age). First, we examined OTR expression in the isocortex (Fig. 3, Supplementary Movies 1–5). Our data showed that overall cortical OTR density reaches its peak at P21 and decreases into adulthood (the red line in Fig. 3b). We also noted spatially heterogeneous expression in different cortical areas (Fig. 3b). For example, the anterior

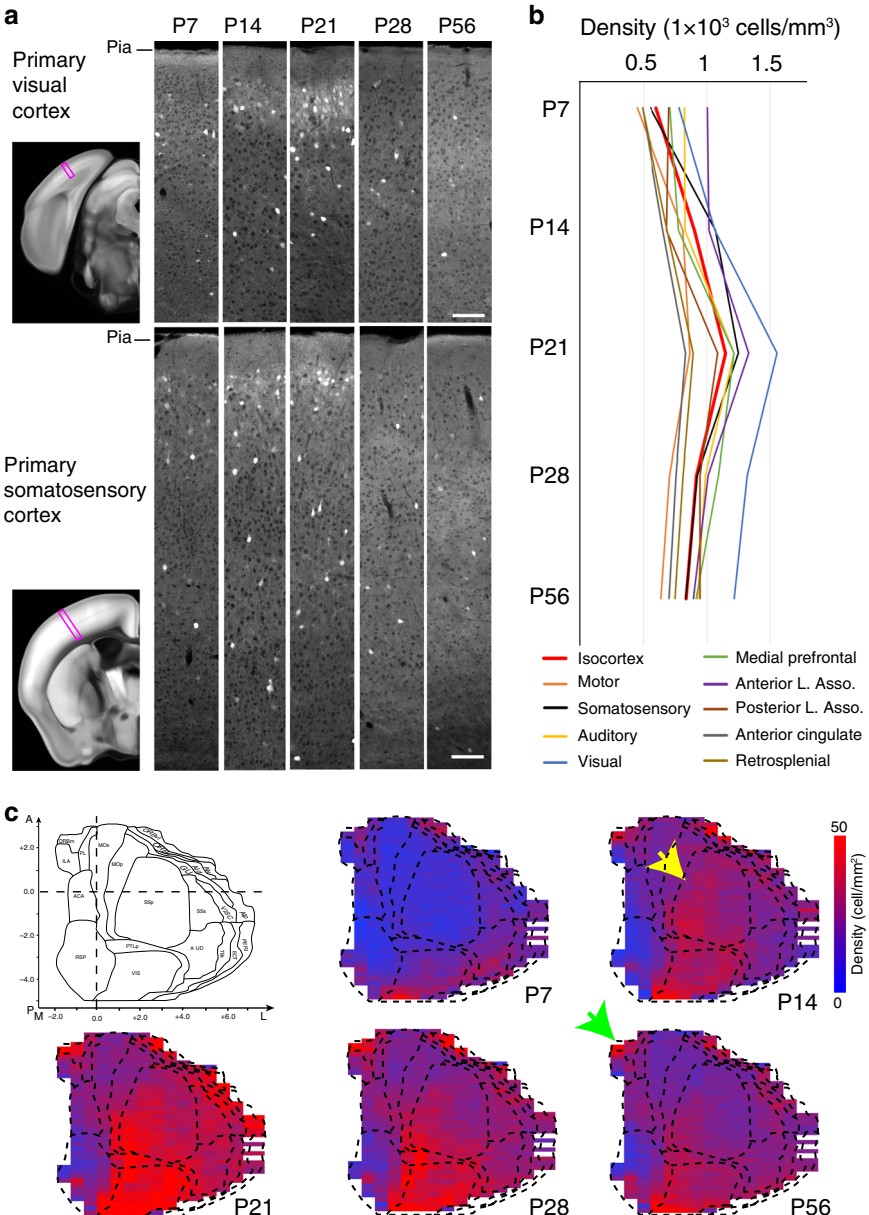

**Fig. 3 Developmental trajectory of OTR cells in the isocortex from *Otr-Venus* mice. a** Representative images from the primary visual and the primary somatosensory cortices (purple boxes) in $Otr^{Venus/+}$ mice at P7, 14, 21, 28, and 56 (columns to the right). Note clustered and dispersed OTR expression in the superficial and deep cortical layers, respectively. Scale bars = 100 μm. **b** Average densities of OTR-Venus cells in different isocortical areas at five different postnatal ages. Anterior lateral (L) association (Asso) area for lateral orbital, gustatory, visceral, and agranular insular; posterior L. Asso. for temporal association, perirhinal, and ectorhinal cortex. See Supplementary Data 1 for more details. **c** 2D cortical flatmap representation of OTR-Venus expression pattern at different developmental time points. The heat map represents OTR-Venus densities in evenly spaced bins in the cortical flatmap. Note overall peak expression in all cortical regions at P21. See also Supplementary Fig. 3 for layer-specific cortical flatmaps. The yellow arrow at P14, and the green arrow at the P56 flatmap highlights the somatosensory cortex and the medial prefrontal cortex, respectively. Full name of abbreviations can be found in Supplementary Data 1.

cingulate and the retrosplenial cortex, parts of the medial association area, showed low OTR density, while the visual and lateral association areas (e.g., the temporal association area) showed higher OTR density (Fig. 3b). Mapping data also revealed temporally heterogeneous patterns. For example, the somatosensory area reached its near peak expression at P14 (bottom panel in Fig. 3a; black line in Fig. 3b), while the visual area showed rapid increases up to P21 (top panel in Fig. 3a; blue line in Fig. 3b). To further visualize the temporally and spatially heterogeneous OTR expression patterns more intuitively, we used cortical flatmaps throughout the developing brain[23]. Cortical flatmaps are digitally

flattened 2D maps of 3D cortical areas that use evenly spaced bins as a spatial unit to quantify and to display detected signals[23]. The cortical flatmap clearly highlighted regional differences in OTR developmental expression with early expression in visual, medial prefrontal, and lateral association area as early as P7 (Fig. 3c). In contrast, somatosensory regions showed little OTR expression at P7 with a rapid increase in OTR density at P14 (yellow arrow in Fig. 3c). By P21, regional heterogeneity attained an adult-like pattern although adults showed lower OTR density overall (Fig. 3c). Interestingly, the medial prefrontal cortex (mPFC) showed a less dramatic decrease in OTR density as mice

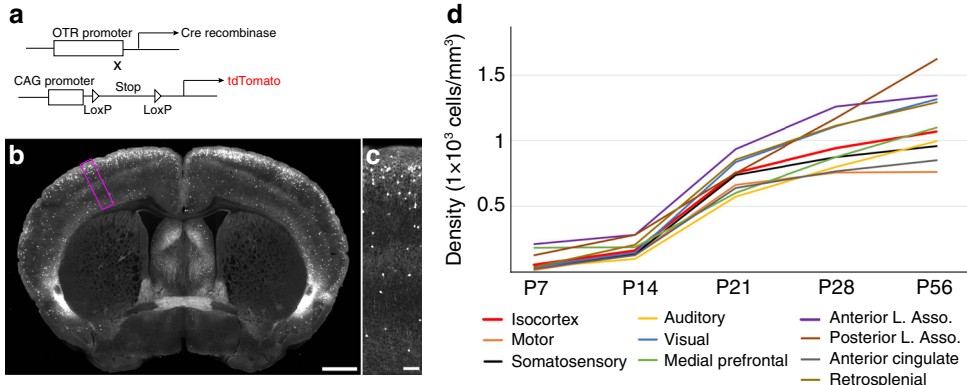

**Fig. 4 OTR downregulation is largely driven by receptor downregulation. a** Experimental design to permanently label transient OTR-positive neurons by crossing *Otr-Cre* with Cre-dependent reporter mice (*Otr-Cre*:Ai14). **b** Example of an adult *Otr-Cre*:Ai14 brain. Scale bar = 1 mm. **c** High magnification image of purple boxed area in **b**. Scale bar = 100 μm. Note the abundant tdTomato-positive cells in the upper layer from the developmental labeling. **d** Average density of tdTomato (+) cells in different isocortical regions at different developmental time points.

progressed to adulthood when compared to other cortical regions, which matches previous results reporting robust OTR expression in the adult mPFC (refs. [14,27]; the green line in the Fig. 3b, the green arrow in 3c).

We also noticed higher OTR density in the superficial cortical layers (layers 1–3) compared to deeper layers (layers 5 and 6), particularly at P14 and P21 (Fig. 3a). To understand how this cortical layer-specific expression affects developmental OTR patterns, we used layer-specific cortical flatmaps to visualize the superficial and deep layer expression patterns separately (Supplementary Fig. 3). We found that OTR in the superficial layer appears earlier and peaks earlier (at around P14) than the deep layers. The superficial layers also show a more pronounced reduction in adulthood compared to the deep layer (Supplementary Fig. 3). At P14, superficial layer expression patterns in the somatosensory cortex match previously reported *Otr* autoradiography and mRNA in situ data (green arrow in Supplementary Fig. 3a)[6,10]. Moreover, relative OTR expression in different cortical layers across postnatal periods clearly showed that OTR expression is predominantly found in the superficial layers at P7 and P14, while older animals have similar or even relatively greater OTR cell density in the deep layer. This pattern is largely driven by the transient OTR expression in the superficial layers (Supplementary Fig. 3c).

Together, these data suggest that developmental OTR expressions differ quantitatively in different cortical areas and even different layers within the same cortical region.

**Receptor downregulation driving postnatal OTR reduction.** Reduction of OTR-expressing cells in the adult isocortex can be explained by either receptor downregulation or programmed cell death during early postnatal development. For example, 40% of interneurons in the mouse cortex are eliminated during the postnatal period via programmed cell death[28]. To understand the main mechanism dictating the transient nature of OTR expression, we crossed mice expressing Cre recombinase under the OTR promotor (*OTR-Cre* knock-in mice) with Cre-dependent reporter mice (Ai14) that express the tdTomato fluorescent reporter (Fig. 4a). The presence of Cre, even if transient as seen during developmental periods, leads to the permanent expression of tdTomato (Fig. 4a). If OTR (+) cells were undergoing cell death, we would expect to see a reduced number of tdTomato (+) cells in the adult brain. On the other hand, if OTR is simply downregulated but the cells remain, tdTomato (+) cell density should not decrease in adulthood. When we quantified cortical tdTomato (+) cells from *Otr-Cre*:Ai14 mice (N = 2 brains for P7, 5 for P14,

6 for P21, 6 for P28, and 6 for P56) using the dqBrain method, we observed that the average density of tdTomato (+) cells began to plateau at around P21 without any reduction in the adult stage at P56 (Fig. 4b–d). Rather, OTR density continued to increase slightly between P21 and P56 largely because of the developmental accumulation of tdTomato within cells leading to slightly higher cell counting in later ages. In summary, this data suggests that developmental regulation of OTR in the isocortex is mainly driven by receptor downregulation, not by cell death.

**Cell type composition in cortical layers.** Neurons in the mouse isocortex are composed of nonoverlapping glutamatergic (excitatory) and GABAergic (inhibitory) neurons with roughly a 4:1 ratio[29]. OTR is known to be expressed in both glutamatergic and GABAergic cortical neurons[14,27,30]. In order to determine the cell type of OTR expressing cortical neurons during postnatal development, we performed immunohistochemistry against GAD67, a marker for GABAergic neurons, in *Otr-Venus* mice at P21 and P56 (N = 3 mice per age, 3 representative sections per brain region; Fig. 5). We examined the mPFC, the somatosensory cortex, and the visual cortex at three different cortical layers; upper layer for layers 1–3, deeper layer for layers 4–6a, and layer 6b (Table 1). We found that the minority of OTR-Venus cells (at ~20%) in both upper and deeper layers are GABAergic in both ages (Fig. 5, Table 1). In contrast, the majority of OTR-Venus cells in the deepest cortical layer (layer 6b) were GABAergic in both ages (Fig. 5, Table 1). Interestingly, a previous study showed that these deep layer OTR-positive neurons are mostly long-range projecting somatostatin neurons[31]. In a separate cohort of *Otr-Venus* mice, we performed fluorescent in situ hybridization with vesicular glutamate transporter 1 (*vGlut1*) as a major marker for excitatory neurons in the cortex at 3 weeks and 10-weeks old (N = 3 mice per age, 2 representative sections per brain region; Supplementary Fig. 4). The majority of OTR-Venus cells in both upper and deeper layers are *Vglut1* positive, while fewer OTR-Venus neurons in the layer 6b co-expresses *Vglut1* (Supplementary Fig. 4, Supplementary Table 1). There was no noticeable difference of OTR neuronal subtype composition in the isocortex between P21 and P56 (Table 1, Supplementary Table 1).

**Developmental expression in subcortical brain regions.** Kinetics of neural circuit maturation vary significantly between different brain regions[32]. Since OTR is also widely expressed in different brain regions outside of the cortex, we sought to find whether these brain regions undergo similar expression trajectory to the isocortex in postnatally developing brains. We first

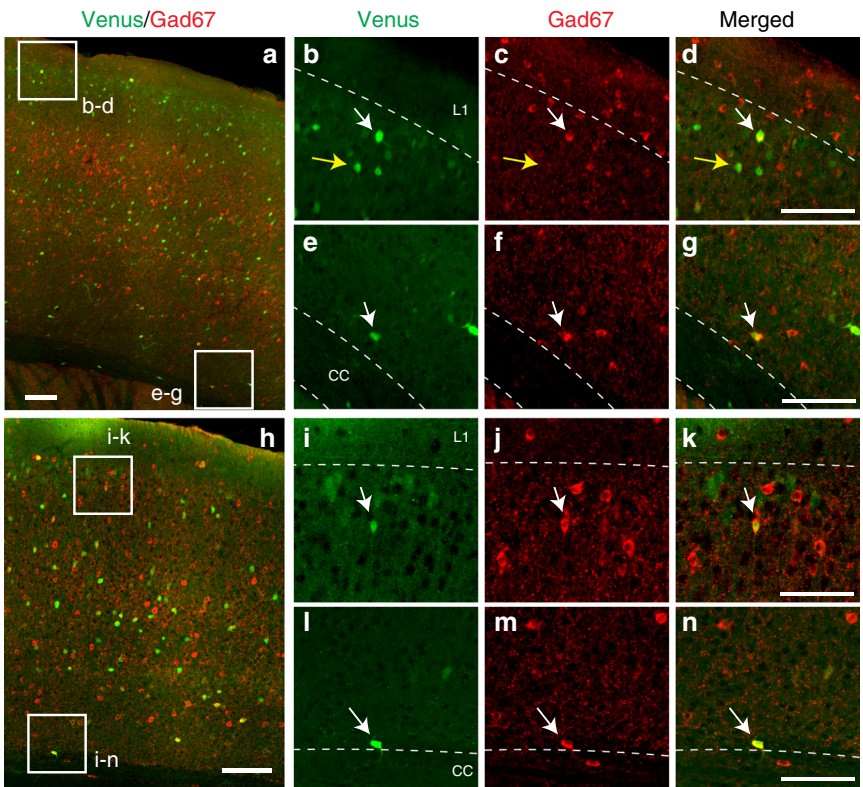

**Fig. 5 OTR cell types in the cortex. a–n** Gad67 antibody immunohistochemistry staining (red) from motor-somatosensory cortical area at around bregma A/P = −0.7 mm from P21 **a**–**g** and P56 **h**–**n** *Otr-Venus* mice (green). Scale bar in **a**, **h** = 100 μm. Examples of high magnification images in the upper layer **b**–**d**, **i**–**k** and the layer 6b **e**–**g**, **l**–**n** from boxed areas in **a**, **h**. Scale bar in **d**, **g**, **k**, **n** = 100 μm. White arrows for examples of Venus (+) cells co-expressing Gad67, and yellow arrows for Venus (+) cells without Gad67 colocalization. L1 in **b**, **i** for the layer 1, cc in **e**, **l** for the corpus callosum.

---

**Table 1 Gad67 colocalization with cortical OTR-Venus neurons.**

| Brain area | P21 | | | P56 | | |
|---|---|---|---|---|---|---|
| | Upper layer | Deeper layer | Layer 6b | Upper layer | Deeper layer | Layer 6b |
| Medial prefrontal cortex | 17% (99/576) | 20% (123/619) | 56% (32/56) | 24% (78/326) | 29% (123/421) | 62% (16/26) |
| Somatosensory cortex | 13% (156/1191) | 19% (301/1581) | 73% (36/49) | 13% (120/927) | 17% (241/1399) | 74% (38/51) |
| Visual cortex | 18% (123/671) | 15% (214/1404) | 92% (43/46) | 19% (131/691) | 19% (342/1809) | 86% (40/46) |

Data from the medial prefrontal cortex (at around Bregma A/P:+1.6), the somatosensory cortex (at around Bregma A/P:−1.0), and the visual cortex (at around Bregma A/P:−3.5). Data presented as percentage of colocalized cells (OTR and Gad-positive cells/total OTR-positive cells) in each brain region.

---

examined ten large brain regions (the olfactory area, the hippocampal area, the striatum, the pallidum, the cerebellum, the thalamus, the hypothalamus, the midbrain, the pons, and the medulla) based on the Allen Brain Atlas ontology[22]. The olfactory areas express OTR at the highest levels (purple line in Fig. 6a, Supplementary Movies 1–5) as exemplified by a very high OTR density in the anterior olfactory nucleus (Fig. 6b). In contrast, the cerebellum and the thalamus showed the lowest OTR densities (gray and yellow lines in Fig. 6a, respectively) with a few noticeable exceptions, including relatively high expression in the paraventricular thalamus (Fig. 6e). There are also several noticeable subcortical areas with strong expression, including the magnocellular nucleus (also called magnocellular preoptic area), a part of the basal forebrain area (Fig. 6c). We also observed prominent expression in specific hippocampal areas, including the subiculum (Fig. 6f). All areas except the hypothalamus reached their peak OTR densities at P21 with slight decrease in adulthood (Fig. 6a). Interestingly, we observed continued increase of OTR in many hypothalamic nuclei until adulthood, including the ventral medial hypothalamus ventral lateral (Fig. 6a, d), which matched

previous OTR binding assays in rats[33]. A detailed list of OTR cell density across all brain regions at different ages can be found in Supplementary Data 1.

**Sexual dimorphism of OTR expression.** OTR is expressed in a sexually dimorphic manner as a part of neural circuit mechanism to generate behavioral differences in males and females[34,35]. Therefore, we compared OTR-Venus expression in male and female mice ($N = 5$ in each male and female brains at different ages) to determine if there were any regions showing strong sexual dimorphism. Across the entire brain region throughout the postnatal development, we found significant sexual dimorphism in two hypothalamic regions (Fig. 7). The ventral premammillary nucleus (PMv) showed significantly higher OTR expression in males compared to females between P14 and P56 (Fig. 7a–d). In contrast, the anteroventral periventricular nucleus (AVPV) near the medial preoptic area showed higher OTR expression in females than males at P56, but not before (Fig. 7e–h). A recent study identified abundant estrogen-dependent OTR-expressing cells in the AVPV, co-expressing estrogen receptor in female

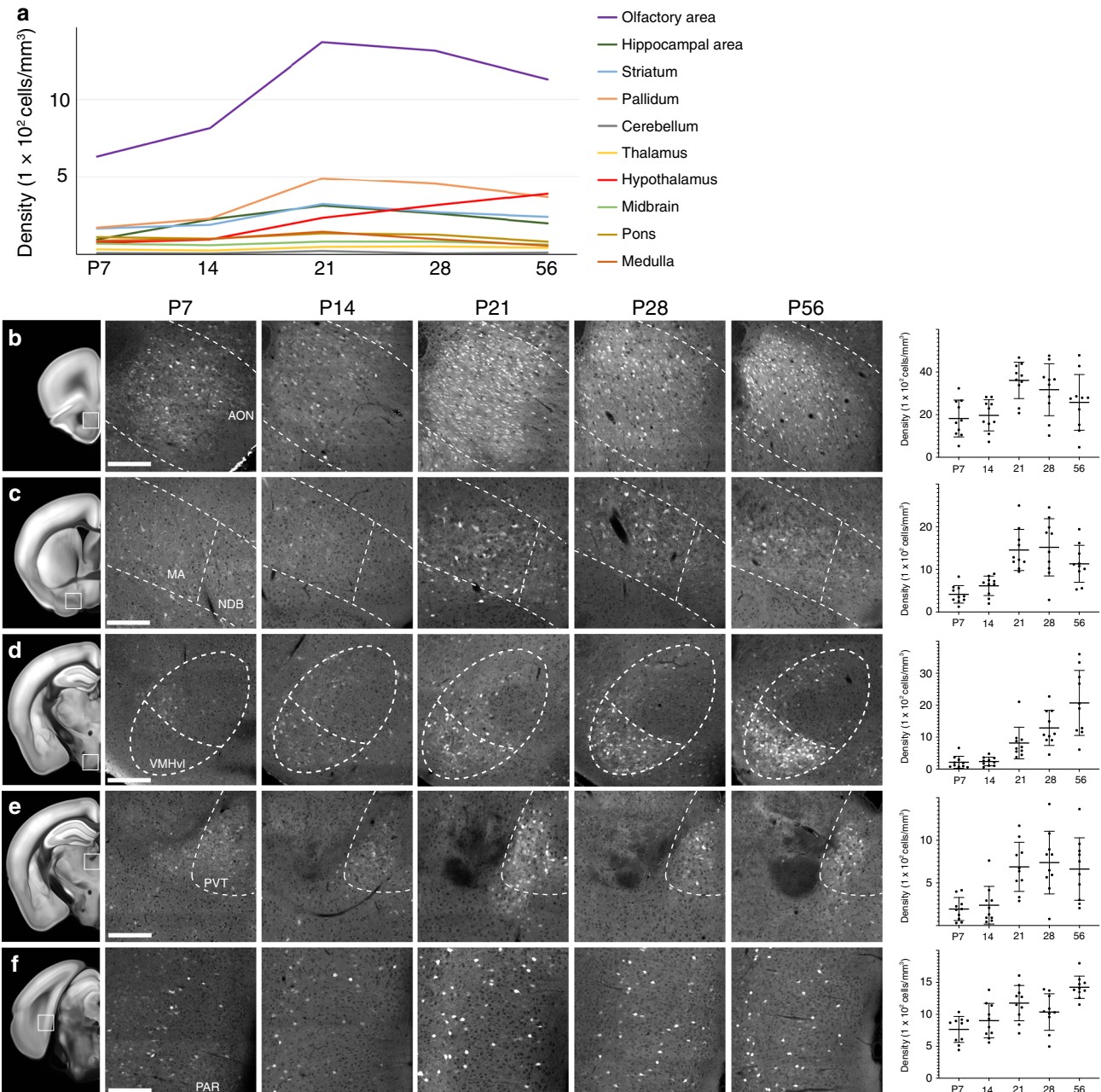

**Fig. 6 Temporal expression patterns in other cortical and subcortical regions. a** Average density of OTR neurons in ten different major subregions of the brain at postnatal development periods. **b–f** Notable brain regions with different temporal expression patterns. The first column highlights anatomical region of interest with white boxes in the adult reference brain. Mid columns represent zoomed-in picture of highlighted brain regions at different ages between P7–P56. Scale bar = 200 µm. The last column is for the OTR (+) cell density measurement of the selected region (error bar = mean ± standard deviation). **b** The anterior olfactory nucleus (AON). **c** The magnocellular regions (MA) and the nucleus of diagonal band (NDB) in the basal forebrain area. **d** The ventral medial hypothalamus ventral medial (VMHvl) in hypothalamic areas. **e** The paraventricular thalamus (PVT) in thalamus. **f** The parasubiculum (PAR) as a part of the retrohippocampal region.

mice[36]. This result suggests a potential role of OTR in sexual behavior[36–38].

**OTR expression in *Cntnap2* KO mouse model of autism.** *Cntnap2* KO mice are a well-established rodent model of autism that recapitulates autism-like behaviors, including impaired social behavior[39]. Reduced OT-producing neurons were observed in *Cntnap2* KO mice and early postnatal OT treatment rescued impaired social behavior in the *Cntnap2* KO mice[40]. Here, we examine whether OTR density is also altered in *Cntnap2* KO mice

by crossing with *Otr-Venus* reporter mice. *Cntnap2* heterozygote mice were used as a control because the heterozygote *Cntnap2* mutation was not associated with autism[41,42]. When densities of OTR expression cells were compared between *Cntnap2* KO ($Otr^{Venus/+}$: $Cntnap2^{-/-}$, $N = 5$ mice) and littermate control ($Otr^{Venus/+}$: $Cntnap2^{+/-}$, $N = 5$ mice) at P21, none of the brain regions showed statistically significant difference (Supplementary Fig. 5). This data suggests that OT signal dysfunction in the *Cntnap2* KO mice is limited to presynaptic reduction of OT without postsynaptic OTR changes.

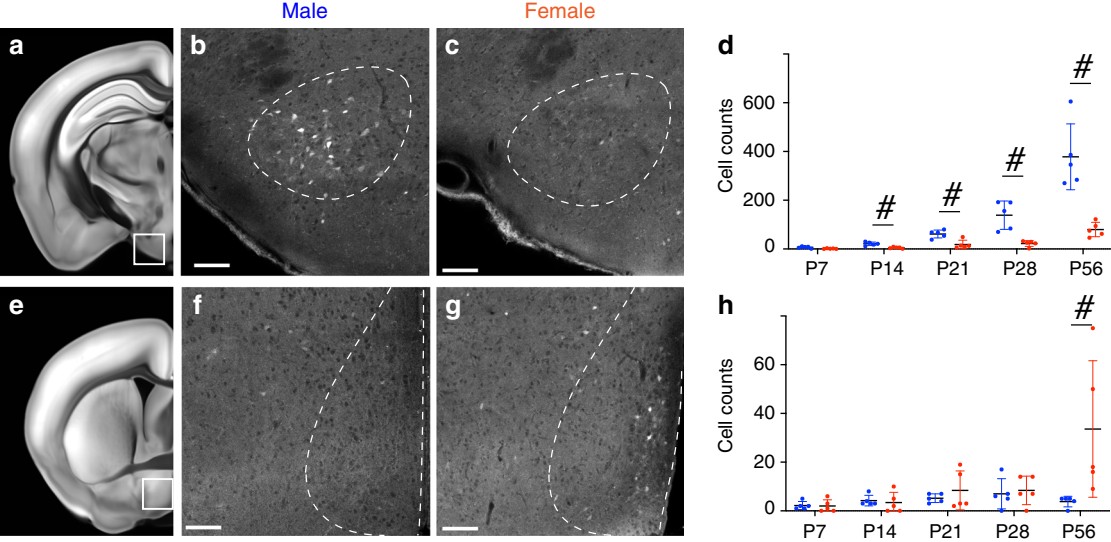

**Fig. 7 Sexually dimorphic expression of OTR neurons. a–d** The PMv showed significantly higher density of OTR cells in males than females from P14. **e–h** In contrast, the AVPV near the medial preoptic nucleus showed significantly higher OTR cells in females than males at P56, but not before. The first column is to highlight anatomical regions of interest in an adult reference brain. The second and the third columns are zoomed-in pictures from P56 adult male and female OTR-Venus brains, respectively. Scale bar in **b**, **c**, **f**, **g** = 100 μm. The last column is number of OTR-Venus (+) cells in the anatomical regions over time (error bars = mean ± standard deviation). Blue dots for males and red dots for females. # Denotes statistically significant difference between males and females with false discovery rate <0.05 after multiple comparison correction.

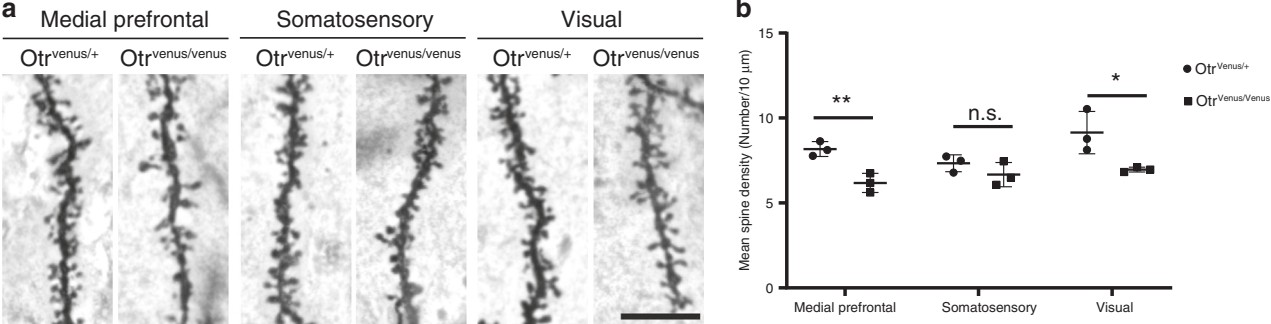

**Fig. 8 Reduced spine density in Otr KO mice. a** Example of Golgi-stained spines in *Otr* heterozygote (*Otr^Venus/+^*) and *Otr* KO (*Otr^Venus/Venus^*) in the medial prefrontal, the somatosensory, and the visual cortex. Scale bar (same across images) = 10 μm. **b** *Otr* KO mice showed significantly reduced spine density in the medial prefrontal and visual cortices compared to *Otr* heterozygote mice. *$p < 0.05$, **$p < 0.01$, n.s. not significant. Error bars = mean ± standard deviation.

**Reduced dendritic spine densities in OTR KO mice.** Previous studies suggest that OT signaling in developing brains plays critical roles in development and function of mature synapses[12,13]. Here, we used Golgi staining to label and compare synaptic densities of *Otr* KO (*Otr^Venus/Venus^*) and heterozygote (*Otr^Venus/+^*) littermate mice at P21 ($N = 3$ mice per genotype). We examined three different cortical areas (medial prefrontal, somatosensory, and visual). Golgi staining showed that the medial prefrontal and visual cortices, but not the somatosensory area, showed significantly reduced dendritic spine density in *Otr* KO mice compared to *Otr* heterozygote mice (Fig. 8). This result suggests that postnatal OTR expression plays a key role in establishing synapses.

**Web-based resource sharing.** Our high-resolution whole brain OTR expression dataset can serve as a resource for future studies examining how OTR regulates different neural circuits in postnatal development and adulthood. Moreover, our newly generated postnatal template brains can be used to map signals from

different studies in the same spatial framework. To facilitate this effort, we created a website (http://kimlab.io/brain-map/OTR) to share our imaging data and other data resources from the current study. Data included in this paper can be easily visualized and downloaded from different web browsers, including mobile devices. We highly encourage readers to explore this whole brain dataset on our website to investigate OTR expression in their regions of interest.

**Discussion**

Here, we provide highly quantitative brain-wide maps of OTR expression in mice during the early postnatal developmental period and adulthood. We establish new mouse brain templates at different postnatal ages and apply our dqBrain method to image and quantify fluorescently labeled signals at cellular resolution in postnatally developing brains[23]. We find spatially and temporally heterogeneous developmental OTR expression patterns in different brain regions that play important roles in the synaptic development. Moreover, we find sexually dimorphic OTR

expression in two hypothalamic regions. Lastly, our high-resolution imaging data is freely accessible via an online viewer as a resource for the neuroscience community.

OT signaling via OTR plays a pivotal role in postnatal brain development and is a key component of multisensory processing required to generate mature social behavior[43,44]. Moreover, quantitative changes of OTR have been correlated with social behavioral variation in both normal and pathological conditions[44,45]. For example, OTR expression levels within a brain region change based on early social experience[46,47]. These findings suggest that OTR expression may be uniquely linked to the early postnatal development of social behavior. Thus, our data provides a quantitative understanding of OTRs developmental patterns in different neural circuits during critical periods of social behavior development.

We chose to use *Otr-Venus* mice to examine the whole brain OTR expression patterns throughout postnatal developmental periods after confirming that this reporter line provides a faithful representation of endogenous OTR expression using fluorescent in situ hybridization. The OTR-Venus expression patterns described here largely agree with previous histological studies focused on selected brain regions and/or ages[6,7,21,33,44,48–51]. With its rapid protein maturation and decay half-life[52,53], Venus served as an ideal reporter protein for developmentally transient OTR expression in the entire brain which enabled us to circumvent laborious histological staining.

Our data driven approach uncovered quantitative insights about postnatal OTR expression. First, there are significantly heterogeneous spatial and temporal patterns of OTR expression across different cortical domains. For example, OTR expression emerged in visual–auditory cortices as early as P7 and propagated to the somatosensory cortex at P14, reaching overall peak expression at P21. Since mice do not open their ears and eyes until about two weeks after birth[54], OTR expression in the visual–auditory areas precedes corresponding sensory inputs. Previous studies showed that OT signaling via OTR promotes synaptogenesis and facilitates synaptic maturation in postnatally developing brains[10,12,55]. We also observed that a lack of OTR negatively impacts dendritic spine density in cortical areas, including the visual cortex. These evidences raise the possibility that early OTR expression may prime cortical areas for incoming sensory signals. Rapid increase of the OTR expression in P14–21 corresponds with the peak time of synaptic formation and maturation in rodent brains[56,57]. Synaptic maturation patterns differ in cortical layers during early postnatal periods[58,59]. For example, tactile stimulus specific activity pattern emerges in the superficial layers of the barrel cortex that is subsequently followed by deep layer maturation in mice[59]. Interestingly, we find that OTR expressed more abundantly in the superficial layers at early postnatal time points (P7 and P14). As mice age, OTR becomes expressed at equal or relatively denser expression in the deep layer. This layer-specific temporal cortical expression is mainly driven by transient OTR expression in the superficial layer at the early postnatal period between P14–21. Since synaptic connections in the superficial layers 2/3 strengthen during this early postnatal period[60], transient OTR is ideally positioned to modulate synaptic maturation in the superficial layer. Second, we found that most subcortical regions also show their peak OTR expression at P21 followed by reduction into adulthood. This pattern agrees with previous observations that adult OTR patterns are established ~3 weeks postnatal age in mice[7]. In contrast, the hypothalamic area showed a continuous increase into adulthood with sexually dimorphic expression of OTR in the PMv and AVPV, parts of the hypothalamic behavioral control column that generates sexually motivated behavior[38]. This suggests that OTR in hypothalamic nuclei plays a role in generating sex-specific behavior during sexual maturation[36,44,61].

From a technical point of view, our dqBrain method is a significant departure from previous semiquantitative histological methods. Our method provides a quantitative way to compare and contrast any fluorescently labeled signals in postnatally developing brains with various experimental conditions. Moreover, our newly established postnatal templates can help to map signals from other 3D imaging modalities (e.g., light sheet fluorescent microscopy) to age-matched spatial framework for quantitative comparisons. Previously, there has been significant effort to create common atlas framework to integrate findings from different studies in the adult mouse brain[62,63]. Our postnatal template brains can serve as a common platform to study various signals from developing brains.

Lastly, our quantitative expression data with easy web-based visualization provides a resource to examine OTR expression of any target brain region at different postnatal ages. Such open data sharing has proven to be useful in disseminating hard-earned anatomical data to the larger scientific community[64,65]. In summary, we envision that our data will guide future circuit-based investigation to understand the mechanism of OT signaling in relationship with different behavioral studies in postnatally developing and adult brains.

## Methods

**Animals**. Animal procedures are approved by Florida State University, Tohoku University, and the Penn State University Institutional Animal Care and Use Committee. Mice were housed under constant temperature and light condition (12 h light and 12 h dark cycle), and received food and water ad libitum. *Otr-eGFP* mice[24] (RRID:MMRRC_012844-UCD) were originally obtained from Mutant Mouse Resource and Research Center (MMRRC) with a mixed FVB/N × Swiss-Webster background strain. *Otr-Venus* mice[20] were originally produced and had their brains collected in Tohoku University (Nishmori Lab). Later, *Otr-Venus* line was imported to the Penn State University (Kim Lab). *Otr-Venus* brains used in the current study came from both Tohoku University and Penn State University. *Otr-Cre* line was originally established by Hidema et al.[66], and imported to the Penn State University via mouse rederivation. Both *Otr-Venus* and *Otr-Cre* mice are 129 × C57BL/6 J mixed genetic background and are not commercially available. *Otr-Cre* mice were then crossed with Ai14 (Jax: 007914, C57Bl/6 J background) to generate *Otr-Cre*:Ai14 mice. For *Otr^{Venus/+}*: *Cntnap2^{−/−}* mice, *Cntnap2^{−/−}* mice (Jax: 017482, C57Bl/6 J background) were initially bred with *Otr^{Venus/+}* mice to create *Otr^{Venus/+}*: *Cntnap2^{+/−}* mice (F1). These F1 mice were further crossed with *Cntnap2^{−/−}* mice to generate *Otr^{Venus/+}*: *Cntnap2^{−/−}* and *Otr^{Venus/+}*: *Cntnap2^{+/−}* (littermate control) mice. All mouse lines were generated using continuously housed breeder pairs and P21 as the standard weaning date.

**Sample preparation, STPT imaging, and related data analysis**. Mice at various postnatal days were perfused by transcardiac perfusion with 0.9% NaCl saline followed by 4% paraformaldehyde in 0.1 M phosphate buffer (PB, pH 7.4). Brains were further fixed overnight at 4 °C and switched to 0.1 M or 0.05 M PB next day until imaging. Fixed brains were embedded in oxidized agarose and cross-linked by 0.05 M sodium borohydride buffer at 4 °C overnight to improve vibratome cutting during STPT imaging. For the STPT imaging, we used 910 nm wavelength laser for *Otr-eGFP* and *Otr-Cre*:Ai14 mice, 970 nm for *Otr^{Venus/+}* mice, respectively. Signals at green and red spectrum were simultaneously collected using 560 nm dichroic mirror. We acquired images at 1 μm (x and y) resolution in every 50 μm z-section throughout the entire brain. Additional information on STPT imaging procedures can be found in previous publications[23,26,62]. Raw data was stitched to reconstruct the brain using custom-built code[23]. For image registration to reference template brains, we used Elastix[67] to register brains to age-matched reference template brains using 3D affine transformation with four resolution level, followed by a 3D B-spline transformation with six resolution level[23]. Image registration parameter for Elastix is included in Supplementary Data 2. We used a machine learning algorithm to detect fluorescently marked cells in serially collected 2D images[23]. To convert the 2D counting to 3D counting, 2D cell counting numbers were multiplied by a 3D conversion factor (1.4) to estimate the total numbers of cells in each anatomical volume based on our previous calculation with cytoplasmic signals[23]. To calculate the volume of each brain region, we registered age-matched template brains to each brain sample using Elastix[67]. Then, voxel numbers of each anatomical label were multiplied by $20 \times 20 \times 50\ \mu m^3$ (3D volume of anatomical voxel unit) to calculate volumes of each anatomical region[23]. The 3D estimates of cell numbers in each anatomical region were divided by corresponding regional volume

to generate density measurement per mm³ in each anatomical area (Supplementary Data 1). All custom-built codes were included in the previous publication[23].

**Statistical analysis**. Density of fluorescently labeled cells in different anatomical regions including flatmap were presented as mean (Figs. 3b, c, 4d and 6a, Supplementary Fig. 3c) or mean ± standard deviation (Figs. 6b–f, 7d, h and 8b). Prism 8 (GraphPad) was used to plot individual data distribution with mean ± standard deviation bar in Figs. 6, 7 and 8, Supplementary Fig. 5. To identify sexual dimorphism and difference between *Cntnap2* genotypes, we performed statistical comparisons between males and females in OTR-Venus cell counts across different anatomical regions using open source statistical package R. We estimated our sample size using the power analysis as performed in our previous publication[23]. When significance level ($\alpha < 0.05$) and assumed effect size (0.85), we expected that over 80% of anatomical regions reach sufficient power with $N = 5$ samples per group. For statistical analysis between groups, we assumed the cell counts at a given anatomical area follow a negative binomial distribution and performed statistical analysis as described before[23,26]. Once the *p* values were calculated, they were adjusted using false discovery rates with the Benjamin-Hochberg procedure to account for multiple comparisons across all ROI locations at each age[23,26]. For dendritic spine density comparison (Fig. 8), spine density was reported as number of spines per 10 µm of dendrite. A Student's *t*-test (two tailed, two-sample equal variance) was used to compare spine density of each region between genotypes.

**Generating reference templates in different postnatal ages**. All the work to generate the reference template brains at different ages was based on 20× down-sized images in *x–y* dimension from the original scale, making each image stack at 20 × 20 × 50 µm (*x, y, z*) voxel resolution. We picked the best-imaged brain with good right–left hemisphere symmetry (designated as a "template brain") at each age and performed image registration using Elastix[67] to register different age-matched brains to the template brain. Then, we averaged the transformed brains after the image registration to generate the averaged template brain at each age (Supplementary Fig. 2). We used either red or green channel images, or both from the same mouse acquired from the STPT imaging. To establish anatomical labels in averaged template brains, we used the image registration method to transform the adult atlas with anatomical labels to fit template brains at different ages. We used the CCF brain and labels from Allen Institute for Brain Science as our initial atlas platform. Direct registration from the adult brain to averaged brains at each postnatal age worked well until the P14 brain due to similarities in postnatal brain morphologies, but not for P7 brains due to more embryonic brain-like shape. To circumvent this issue, we registered the adult CCF to the P14 template brain first, then the transformed CCF fit to the P14 brain was registered again to P7 (Supplementary Fig. 2). This sequential registration worked because the morphological difference between P7 and P14 was smaller than difference between P7 and the adult brain.

**Cortical flatmap**. We previously generated evenly spaced cortical bins to generate a cortical flatmap in an adult reference brain and devised a method to map detected signal in the flatmap[23]. Here, we further generated superficial and deep layer cortical flatmaps. First, we created a binary file with layers 1–3 for superficial and layers 5 and 6 for deep layer across the entire isocortex. Second, we used the binary filter to remove unwanted cortical areas from the existing isocortical flatmap in order to create layer-specific cortical flatmap. To quantify signals on flatmaps, we registered all samples to the reference brain with cortical area bins using Elastix and quantified target signals in each cortical bin, as described in the STPT related data analysis above. We also performed reverse image registration to warp the adult reference brain to postnatal template brains in order to calculate the area of cortical bins at different ages. Then, we calculated densities in each cortical bin based on number of cells and area measurement in each bin. Lastly, the density was plotted in the cortical flatmap using Excel (Microsoft) and Adobe illustrator as described before[23].

**Single-molecule mRNA fluorescence in situ hybridization**. Mice were deeply anesthetized using intraperitoneal injection of anesthesia (100 mg kg⁻¹ ketamine mixed with 10 mg kg⁻¹ xylazine). Then, the animal was decapitated with scissors, and the brain was immediately dissected out and immersed in optimal cutting temperature media (Tissue-Tek). The immersed brain was quickly frozen using dry ice chilled 2-methylbutane. The frozen brain was stored at −80 °C until used. A cryostat was used to collect coronal brain sections at 10 µm thickness. Sections were stored at −80 °C, and in situ hybridization was performed within 2 weeks of sectioning. We used RNAScope detection kits (ACDBio) to detect and to quantify target mRNA at single-molecule resolution. We followed the manufacturer's protocols with the exception that protease III (ACDbio, cat. no. 322340) was applied to tissue for 20 min. Probe-mm-Venus-C1 (ACDbio, cat. no. 493891) and Probe-mm-OTR-C2 (ACDbio, cat. no. 402651-C2) were used to detect *Venus* and *Otr*, respectively. Amp4 Alt A was used to detect *Otr* alone in red channel, and Alt C was applied to detect *Otr*, and *Venus* in far red and red channel, respectively. To quantify colocalization of *Otr-Venus*-positive cells with markers for excitatory neurons, Probe-mm-slc17a7-C2 (ACDbio, cat. no. 416631-C2) and Probe-mm-Venus-C1 (ACDbio, cat. no. 493891) were used to detect mRNA expression of

*Venus* and *Vglut1* mRNA. Amp4 Alt C was used to detect *Vglut1* and *Venus* in the far red and red channels, respectively.

**Immunohistochemistry**. *Sample preparation*: *Otr*$^{Venus/+}$ mice of both sexes were collected at P21 and P56. Mice were deeply anesthetized with intraperitoneal injection of the ketamine/xylazine mixture. Then, mice were transcardially perfused with 0.9% NaCl saline followed by 4% PFA. Whole heads were removed and post-fixed in the same fixative at 4 °C for 3 days. Then, the brain was dissected out and sunk down in 30% sucrose in 1× PBS (pH 7.4) solution at 4 °C for cryoprotection. Cryoprotected brains were then frozen on dry ice and stored at −20 °C until sectioning. A total of 30 µm thick coronal sections were obtained using a freezing microtome (Leica). Sections were stored in a cryoprotectant solution (30% sucrose and 30% glycerol in 0.1 M PB) at −20°C until immunostaining.

*Immunostaining*: All washes were performed for 10 min at room temperature with gentle rotation unless otherwise specified. Free floating sections were washed in 1× PBS three times followed by 1 h of blocking with 1% donkey serum diluted in 1× PBS at room temperature. Slices were then incubated with a monoclonal primary antibody (mouse anti-GAD67, Millipore Cat# MAB5406, RRID: AB_2278725, diluted 1:2000) in blocking buffer overnight at 4 °C with gentle rotation. Following primary antibody incubation, the slices were washed in 1× PBS three times and incubated with secondary antibody (Donkey anti-mouse conjugated with Alexa 568, Thermo Fisher Scientific Cat# A10037, RRID: AB_2534013, diluted 1:500) for 1 h at room temperature. Three washes were performed in 0.05 M PB prior to mounting slices with vectashield mounting media (vector laboratories, cat. no. H-1500-10).

**Golgi staining**. To quantify the number of dendritic spines, modified Golgi–Cox staining was performed as described by Bayram-Weston Z et al.[68].

*Sample Preparation*: *Otr*$^{Venus/+}$ and *Otr*$^{Venus/Venus}$ mice of both sexes were collected at P21. Mice were deeply anesthetized with intraperitoneal injection of a ketamine/xylazine mixture. Then, mice were transcardially perfused with 0.9% NaCl saline followed by 4% PFA. Whole brains were dissected out and post-fixed in the same fixative at 4 °C overnight. The brains were then stored in 0.05 M PB until staining was performed.

*Golgi staining*: Brain hemispheres were separated and immersed in Golgi solution for 5 days at room temperature. To make Golgi staining solution, 80 ml of 5% potassium chromate (Sigma-Aldrich, cat. no. P5271) was diluted with 200 ml of diH₂O. Then 200 ml of 2.5% mercuric chloride (Fisher, cat. no. M1136) and 2.5% of potassium dichromate (Sigma-Aldrich, cat. no. 216615) was added to the solution. The final solution was filtered out and stored in the dark at room temperature. The stained brains were sunk down in 30% sucrose in 1× PBS (pH 7.4) solution at 4 °C for 24 h. After 24 h of sinking the brains were transferred to fresh 30% sucrose solution and stored at 4 °C for 48 h. Cryoprotected brains were frozen on dry ice. A total of 80 µm thick coronal sections were obtained using a freezing microtome (Leica). Slices were stored in 0.05 M PB at 4 °C until counter staining.

*Counter staining*: A total of 80 µm thick sections from Golgi-stained brains were mounted on gelatin coated slides and dried for 15 min at room temperature. Then slides were subjected to the following washes: distilled H₂O for 2 min, 20% ammonium for 10 min, distilled H₂O for 2 min, 70% ethanol for 5 min, 90% ethanol for 5 min, 100% ethanol for 5 min, xylene for 10 min, and a second xylene wash for 10 min. The slides were then removed from the final wash and air dried before coverslipping with Eukitt quick hardening media (Sigma-Aldrich, cat. no. 03989).

**Microscopic imaging and quantification**. For both RNA in situ and immunostaining, BZ-X700 fluorescence microscope (Keyence) was used to image large areas using 20× objective lens with 2D image tile stitching. The sectioning function provided a deconvolution mechanism to capture sharply focused images. For *Venus/Vglut1* RNA in situ, BZ-X700 fluorescence microscope (Keyence) was used to image large areas using a 40× objective lens with 2D image tile stitching. For Golgi staining, BZ-X700 fluorescence microscope (Keyence) was used to image large areas using a 60× oil immersion objective lens with 2D image tile stitching. Images with large field of view were exported as Tiff files using the BZ-X analyzer software (Keyence). Image evaluation and cell counting was performed manually using the cell counter plug-in in FIJI (ImageJ, NIH)[69]. OTR/GAD and Venus/OTR cell counting was done blindly by two independent experts. For the *Venus* and *Otr* colocalization in Fig. 1v, *Venus* with more than four puncta from the RNA in situ was considered a Venus (+) cell. Images were acquired from cortical, amygdala, and hippocampal regions. In both OTR-Venus and OTR-Gad67 colocalization studies, two experts agreed over 95% of colocalization assessment. The final reported number is the averaged value from two expert's counting. For *Venus/Vglut1* in situ counting, one expert blindly counted cells containing at least four puncta Venus puncta were considered positive. Within this population, cells were considered *Vglut1* positive if they contained four puncta. For Golgi spine quantification, one expert used the simple neurite tracker FIJI plug-in (imageJ, NIH)[69] to measure the length of dendrite then the cell counter plug-in (imageJ, NIH)[69] to quantify spines along the traced dendrites without knowing genotypes of mice.

**Reporting summary**. Further information on research design is available in the Nature Research Reporting Summary linked to this article.

## Data availability

Template brains and associated anatomical labels at different postnatal development is available as Supplementary Data 3. Layer-specific cortical flatmap label files are available as Supplementary Data 4. Representative high-resolution images of both *Otr-Venus* and *Otr-eGFP* mice can be found in http://kimlab.io/brain-map/OTR/. All full-resolution images of *Otr-Venus*, *Otr-eGFP*, and *Otr-Cre:Ai14* mice are freely available to download in the Brain Image Library at ftp://download.brainimagelibrary.org:8811/56/fb/56fb1b25ca6b5fae/.

## Code availability

Custom-built codes to reconstruct images from serial two-photon tomography, to detect signals, to quantify signals in reference brains, and to plot signals in cortical flatmaps were publicly distributed in Kim et al.[23] Elastix image registration parameter files can be found in Supplementary Data 2. All codes can be used without any restriction.

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

## Acknowledgements
This publication was made possible by a NIH R01 MH116176 and Tobacco Cure Funds from the Pennsylvania Department of Health to Y.K., Strategic Research Program for Brain Sciences from Japan Agency for Medical Research and Development (AMED; 18dm0107076h0003, 2016–2020) to K.N. and S.H., JSPS Grant-in-Aid for Scientific Research (15H02442, 2015–2018) to K.N., NIH R01 MH114994 to E.A.D.H., and T32 DC000044 to M.T. We thank Rebecca Betty for assistance in editing the manuscript and computational resources from High Performance Computing cluster in Penn State College of Medicine. Its contents are solely the responsibility of the authors and do not necessarily represent the views of the funding agency.

## Author contributions
Project conceptualization: Y.K.; brain sample preparation and data acquisition: K.T.N., Z.T.N., U.C., M.T., S.H., K.N., and E.A.D.H.; data analysis: K.T.N., Z.T.N., A.R.W., and Y.K.; web visualization: D.J.V.; and manuscript preparation: K.T.N. and Y.K. with help of other authors.

## Competing interests
The authors declare no competing interests.
