## [Peer Review File · Nature Communications]

Reviewers' Comments:

Reviewer #1:

Remarks to the Author:

Newmaster and colleagues provided a novel template of mouse developing brains at postnatal day 7, 14, 21, and 28 with detailed anatomical labels through the application of their quantitative brain mapping platform. Using this method in combination with oxytocin receptor (OTR)-Venus knock-in mice, the authors investigated the distribution of OTR-expressing cells in whole brains at different postnatal developmental periods and clearly showed the spatial and temporal heterogeneity of OTR expressions in cortical and subcortical regions. The basic findings in the current manuscript can be summarized as follows: 1) OTR-Venus knock-in mice specifically report Venus expression in OTR-expressing cells proven by in situ hybridization, 2) overall cortical OTR density reaches its peak at postnatal day 21 and decreases into adulthood, 3) OTR density in the superficial cortical layers peaks earlier compared with deeper layers, 4) reduction of OTR-expressing cells into adulthood is caused by receptor downregulation, not by programmed cell death, and 5) there is significant sexual dimorphism of OTR expression in the ventral premammillary nucleus and anteroventral periventricular nucleus.

The data presented in the manuscript is objective and easy to understand intuitively because of their state-of-art method for quantitative brain mapping.

However, my major concern is that the impact and novelty of the reported findings are relatively insufficient due to the lack of functional analysis.

Although the OTR expression in postnatally developing brains has been reported to play an important role in neural circuit maturation of various modalities so far, it remains to be elucidated whether the precise regulation of OTR expression in developing brains is also responsible for neural connectivity. It would be valuable to address this interesting question, although it might be beyond the authors' interest.

Minor comments are following:

1. For further characterization of OTR-expressing cells in the cortex, immunohistochemistry against a marker for glutamatergic cortical neurons would be also helpful besides for GABAergic neurons.
2. In Figure 1, the authors should put detailed information regarding images in figure itself so that readers can easily understand what respective images represent.

Reviewer #2:

Remarks to the Author:

In this report, Newmanster et al describe a developmental quantitative 3D brain atlas of oxytocin receptor (OTR) in mice. The newly developed postnatal developmental mouse brain atlas is impressive, accompanied by informatics tools for quantifying and mapping anatomical data. The manuscript is well written and all the figures are in good quality. Overall, I think this paper is already in good shape for publishing.

Reviewers' comments:

Reviewer #1 (Remarks to the Author):

Newmaster and colleagues provided a novel template of mouse developing brains at postnatal day 7, 14, 21, and 28 with detailed anatomical labels through the application of their quantitative brain mapping platform. Using this method in combination with oxytocin receptor (OTR)-Venus knock-in mice, the authors investigated the distribution of OTR-expressing cells in whole brains at different postnatal developmental periods and clearly showed the spatial and temporal heterogeneity of OTR expressions in cortical and subcortical regions. The basic findings in the current manuscript can be summarized as follows: 1) OTR-Venus knock-in mice specifically report Venus expression in OTR-expressing cells proven by in situ hybridization, 2) overall cortical OTR density reaches its peak at postnatal day 21 and decreases into adulthood, 3) OTR density in the superficial cortical layers peaks earlier compared with deeper layers, 4) reduction of OTR-expressing cells into adulthood is caused by receptor downregulation, not by programmed cell death, and 5) there is significant sexual dimorphism of OTR expression in the ventral premammillary nucleus and anteroventral periventricular nucleus.

We appreciate the reviewer's accurate summary of our manuscript

The data presented in the manuscript is objective and easy to understand intuitively because of their state-of-art method for quantitative brain mapping.

Thank you

However, my major concern is that the impact and novelty of the reported findings are relatively insufficient due to the lack of functional analysis.

Although the OTR expression in postnatally developing brains has been reported to play an important role in neural circuit maturation of various modalities so far, it remains to be elucidated whether the precise regulation of OTR expression in developing brains is also responsible for neural connectivity. It would be valuable to address this interesting question, although it might be beyond the authors' interest.

We performed two additional experiments to address the reviewer's concern.

1. Golgi staining to examine number of synapses in Otr Knockout (KO) mice ($Otr^{Venus/Venus}$) in comparison to *Otr* heterozygote littermate ($Otr^{Venus/+}$) mice. We observed significantly reduced dendritic spine density in the medial prefrontal and visual cortices, but not somatosensory cortex, from *Otr* KO mice. We added following paragraph in the result section with a new Figure 8 and updated other parts of the manuscript accordingly.

Reduced dendritic spine densities in OTR knockout mice

Previous studies suggest that oxytocin signaling in developing brains plays critical roles in development and function of mature synapses^{13,43}. Here, we used Golgi staining to label and compare synaptic densities of *Otr* knockout (*Otr*^{Venus/Venus}) and heterozygote (*Otr*^{Venus/+}) littermate mice at P21 (N = 3 mice per genotype). We examined three different cortical areas (medial prefrontal, somatosensory, and visual). Golgi staining showed that the medial prefrontal and the visual cortices, but not the somatosensory area, showed significantly reduced dendritic spine density in *Otr* knockout mice compared to *Otr* heterozygote mice (Figure 8). This result suggests that postnatal OTR expression plays a key role in establishing synapses.

2. Reduced oxytocin producing neurons have been reported in *Cntnap2* KO mouse model of autism before. Yet, it remains unknown whether oxytocin receptor expressing neurons are also decreased in the *Cntnap2* KO mice. We crossed *Cntnap2* KO mice with OTR-Venus mice and compared OTR densities between *Cntnap2* KO and *Cntnap2* heterozygote littermate control our using dqBrain method. We did not observe statistically significant differences in any brain regions. The following paragraph was added in the results section with a new Supplementary Figure 5.

OTR expression in *Cntnap2* knockout mouse model of autism

Cntnap2 knockout (KO) mice are a well-established rodent model of autism that recapitulates autism-like behaviors including impaired social behavior³⁹. Reduced oxytocin producing neurons were observed in *Cntnap2* KO mice and early postnatal oxytocin treatment rescued impaired social behavior in the *Cntnap2* KO mice⁴⁰. Here, we examine whether oxytocin receptor density is also altered in *Cntnap2* KO mice by crossing with *Otr-Venus* reporter mice. *Cntnap2* heterozygote mice were used as a control because the heterozygote *Cntnap2* mutation was not associated with autism^{41,42}. When densities of oxytocin receptor expression cells were compared between *Cntnap2* KO (*Otr*^{Venus/+}; *Cntnap2*^{-/-}, N = 5 mice) and littermate control (*Otr*^{Venus/+}; *Cntnap2*^{+/-}, N = 5 mice) at P21, none of the brain regions showed statistically significant difference (Supplementary Figure 5). This data suggests that oxytocin signal dysfunction in the *Cntnap2* KO mice is limited to presynaptic reduction of oxytocin without postsynaptic OTR changes.

Minor comments are following:

1. For further characterization of OTR-expressing cells in the cortex, immunohistochemistry against a marker for glutamatergic cortical neurons would be also helpful besides for GABAergic neurons.

We tried CamKII antibody staining to label glutamatergic neurons in the cortex using two different antibodies; CAMKII Alpha Monoclonal 6G9 (cat.no.: MA1048) from Thermo Fisher and Rabbit Anti-CaMKII alpha antibody (ab92332) from Abcam. Unfortunately, none of the antibodies resulted in reliable and high quality of staining.

Instead, we performed fluorescent in situ hybridization method (RNAscope) to detect vGlut1 mRNA as this is the major glutamatergic marker in the cortex. Although a small fraction of cortical excitatory neurons is known to express vGlut2, we observed that almost all vGlut2 positive cells also expressed vGlut1 in the cortex. Colocalization of vGlut1 and OTR-Venus positive neurons showed that majority of OTR neurons in the cortex are glutamatergic except ones in the layer 6b.

The quantification was added in the Supplementary Table 1.

The following text has been added in the result section.

In a separate cohort of *Otr-Venus* mice, we performed fluorescent *in situ* hybridization with vesicular glutamate transporter 1 (*vGlut1*) as a major marker for excitatory neurons in the cortex at P21 and 10 weeks old (N = 3 mice per age, 2 representative sections per brain region; Supplementary Figure 4). The majority of OTR-Venus cells in both upper and deeper layers are *Vglut1* positive while lesser percentage of OTR-Venus neurons in the layer 6b co-expresses *Vglut1* (Supplementary Figure 4, Supplementary Table 1).

2. In Figure 1, the authors should put detailed information regarding images in figure itself so that readers can easily understand what respective images represent.

We added more captions inside the figure to make it more intuitive.

Reviewer #2 (Remarks to the Author):

In this report, Newmanster et al describe a developmental quantitative 3D brain atlas of oxytocin receptor (OTR) in mice. The newly developed postnatal developmental mouse brain atlas is impressive, accompanied by informatics tools for quantifying and mapping anatomical data. The manuscript is well written and all the figures are in good quality. Overall, I think this paper is already in good shape for publishing.

Thank you very much.

Reviewers' Comments:

Reviewer #1:

Remarks to the Author:

The authors significantly improved the manuscript adding the animal models of ASD. I recommend to accept the manuscript for publication.

Reviewer #2:

Remarks to the Author:

The authors have properly addressed Reviewer 1's comments. I do not have any concerns. This is a fantastic paper. I totally support its publication in Nature Communications.

Reviewers' comments:

Reviewer #1 (Remarks to the Author):

The authors significantly improved the manuscript adding the animal models of ASD. I recommend to accept the manuscript for publication.

Thank you

Reviewer #2 (Remarks to the Author):

The authors have properly addressed Reviewer 1's comments. I do not have any concerns. This is a fantastic paper. I totally support its publication in Nature Communications.

Thank you